# Standardising Pronunciation for a Grapheme-to-Phoneme Converter for Faroese

**Sandra Saxov Lamhauge[1], Iben Nyholm Debess[1], Carlos Daniel Hernández Mena[2], Annika Simonsen[3], Jon Gudnason[2]**

[1]The University of the Faroe Islands
[2]Reykjavík University
[3]The University of Iceland

sandrasl@setur.fo, ibennd@setur.fo, carlosm@ru.is, annika@hi.is, jg@ru.is

## Abstract

Pronunciation dictionaries allow computational modelling of the pronunciation of words in a certain language and are widely used in speech technologies, especially in the fields of speech recognition and synthesis. On the other hand, a grapheme-to-phoneme tool is a generalization of a pronunciation dictionary that is not limited to a given and finite vocabulary. In this paper, we present a set of standardized phonological rules for the Faroese language; we introduce FARSAMPA, a machine-readable character set suitable for phonetic transcription of Faroese, and we present a set of grapheme-to-phoneme models for Faroese, which are publicly available and shared under a creative commons license. We present the G2P converter and evaluate the performance. The evaluation shows reliable results that demonstrate the quality of the data.

## 1 Introduction

Pronunciation dictionaries are important components in speech technology (i.e. for ASR and TTS). They are used to link orthographic forms with their pronunciations. There are many kinds of pronunciation dictionaries; some that only provide a standard pronunciation (Weide, 1998), and some that also provide dialectal variants (Robinson, BEEP dictionary). However, dictionaries are always going to be limited to their entries; it is therefore that one would train grapheme-to-phoneme (G2P) models on pronunciation dictionaries and use those models to automatically transcribe unknown words (Nikulásdóttir et al., 2018).

In recent years, there has been steady progress in Faroese speech technology, and the work is ongoing (e.g. Ingason et al., 2012; Hernández Mena et al., 2022a; 2022b; 2022c). A Faroese pronunciation dictionary was created for the first Faroese ASR project, Ravnur and published in 2022 as part of a Basic Language Resource Kit (BLARK) for Faroese (Debess et al., 2022; Simonsen et al., 2022)[1]. A BLARK is defined as the minimal collection of language resources needed to develop language technology for a specific language (Krauwer, 2003; Maegaard et al., 2006). However, the BLARK is not limited to ASR, but can be used to develop a wide range of LT for Faroese.

The dictionary from the Ravnur project uses the variation spoken in the capital of the Faroe Islands as a standard (an arbitrary choice as discussed in Section 5). When creating the grapheme-to-phoneme (G2P) models, it was decided to expand the dictionary into two different pronunciation dictionaries based on dialectal difference, one CENTRAL and one EAST. This way, we could cover more ground and open up the possibility to expand the work further in the future.

Because this was the first open source Faroese dictionary of its kind, Ravnur also had to create a Faroese SAMPA alphabet (called FARSAMPA here) to use for the pronunciations. The process and decisions made when creating FARSAMPA

---

[1] The BLARK for Faroese is open source, published under a CC BY 4.0 licence on the platform OpenSLR (Debess et al., 2022).

and the Faroese pronunciation dictionary have never been published until now. We therefore set out to describe the work that went behind creating these fundamental language resources and how we adapted them to create a reliable G2P model.

In Section 2, we present FARSAMPA. Section 3 outlines some of the most common grapheme-to-phoneme conversions of Faroese, while Section 4 presents the standardization of phonetic variation for the G2P. In Section 5, we discuss Faroese dialectal variation and the decision to make two different pronunciation dictionaries to train G2P models on. Finally, we introduce our G2P models that have been trained on the two pronunciation dictionaries, present an evaluation, and conclude.

| Consonants | FARSAMPA | IPA | Vowels | FARSAMPA | IPA |
|---|---|---|---|---|---|
| Stops | p | pʰ | Monophthongs | i | i |
| | b | p | | I | ɪ |
| | t | tʰ | | e | e |
| | d | t | | E | ɛ |
| | k | kʰ | | a | a |
| | g | k | | y | y |
| Fricatives | f | f | | Y | ʏ |
| | v | v | | 2 | ø |
| | 4 | ð | | 9 | œ |
| | 5 | θ | | u | u |
| | s | s | | U | ʊ |
| | S | ʃ | | o | o |
| | z | ş | | O | ɔ |
| | h | h | | 8 | ə |
| Affricates | tS | tʃʰ | Diphthongs | EA | ɛa |
| | dZ | tʃ | | OA | ɔa |
| Nasals | m | m | | UJ | ʊi |
| | M | m̥ | | EJ | ɛi |
| | n | n | | aJ | ai |
| | x | ņ | | aW | au |
| | N | ŋ | | OJ | ɔi |
| | X | ŋ̊ | | OW | ɔu |
| Laterals | l | l | | 3W | ʉu |
| | L | ļ | | EW | ɛu |
| Approximants | w | w | | 9W | œu |
| | j | j | | 9J | œi |
| | r | ɹ | Diacritics | H | ʰ |

Table **1**: Overview of FARSAMPA mapped to IPA-symbols[2].

## 2 FARSAMPA – Faroese SAMPA[3]

This section introduces FARSAMPA: a machine-readable character set suitable for phonetic transcription of Faroese. This character set is a phonetic alphabet consisting of an inventory of ASCII symbols mapped onto symbols of IPA (the International Phonetic Alphabet). The inventory content is based on Faroese phonetic and phonological knowledge and research (e.g. Petersen, 2021; Thráinsson et al., 2012), and transcription conventions for Faroese were considered when creating the alphabet. All phonemes in the G2P tool introduced in this article are written in FARSAMPA, as the lexical data of the tool is transcribed in FARSAMPA. See Table 1 for an overview of the inventory and characters in FARSAMPA. FARSAMPA was developed as a part of the BLARK in the Ravnur project. The alphabet covers all Faroese phonemes and a few allophones. The alphabet was used for transcribing all words in the Ravnur lexicon (350.000 word forms) and has thereby been tested, adjusted, and proved suitable and sufficient for Faroese phonetic transcription when working with language technology. Every character in the alphabet is directly translatable to an IPA character, making all transcriptions readily convertible to IPA or other systems mapped up against IPA.

The primary purpose of FARSAMPA is to make phonetic transcriptions machine-readable and ready for automatic processing, and even though the value of characters is arbitrary to a machine, we need to keep in mind that phonetic transcription also entails a manual element and needs to be somewhat human-readable as well.

The designation of characters to phones was based on simplicity, efficiency, and intuition. The basic guidelines[4] are summed up as follows:

- All IPA symbols that coincide with a lowercase letter from the latin alphabet are designated the same character in FARSAMPA (e.g. 'p', 'f', 'o').

---

[2] Note that the FARSAMPA also includes length distinction, primary and secondary stress (:, %, ~), but these suprasegmental attributes have not been used in the phonetic transcriptions of the words in the data set for this project making the G2P-tool and will therefore not be introduced further at this point.

[3] The use of the term SAMPA stems from this project: https://www.phon.ucl.ac.uk/home/sampa/. The Faroese

SAMPA from the Ravnur project is not originally published under the name FARSAMPA, but we will use FARSAMPA to refer to the Faroese SAMPA in this article.

[4] Based on the original SAMPA recommendations for Danish, Dutch, English, French, German and Italian, https://www.phon.ucl.ac.uk/home/sampa/.

- Categorical similarities, e.g. 'O / ɔ' and 'N / ŋ' (based on standard SAMPA guidelines).
- Graphic similarities, e.g., 'S / ʃ' and 'U / ʊ'.
- With no letter available, numbers are used over other keyboard symbols as they are easily accessible on qwerty-keyboards language wide and easier to name.
- Avoid designating counterintuitive characters, e.g. 'T' for a vowel. In those cases, numbers or other symbols are preferable.

Not all allophones are represented, as that level of detail does not benefit accuracy or efficiency of language technology purposes. Four unvoiced allophones of the sonorant phonemes /m, n, N, l/ are included, as they are acoustically very different from their voiced counterparts. The only diacritic included with its own character is the preaspiration /H/. Postaspiration and other relevant diacritics for Faroese (e.g. voiced/unvoiced) are integrated in the character system and not denoted via symbols of their own (e.g. [p] and [b] only differ by postaspiration, [m] and [M] only differ by voicing).

## 3 Phonological rules of Faroese

In this section, we will outline some of the most common grapheme-to-phoneme conversions of (central) Faroese. For dialectal variation, see Section 5. The conversions are based on Thráinsson et al. (2012). Almost all of the rules listed have exceptions, and there are other phonological rules as well. For a more thorough overview, see Thráinsson et al. (2012). Furthermore, the phonetics of Faroese are generally understudied (Lamhauge, 2022), and therefore, many phonological conditions have not been sufficiently studied or described. Some of these phonological conditions will be discussed as well.

### 3.1 Grapheme-to-phoneme conversions

In native Faroese words, stressed vowels are short when followed by two or more consonants, except after the following consonant clusters: *pr, pl, tj, tr, kj, kr, kl, sj*[5]. Stressed vowels are long in all other positions. Table 3 (next page) shows the

different graphemes representing vowels in Faroese and their phonemic counterparts.

Table 2 shows the most frequent pronunciation of the consonants in Faroese. In certain grapheme combinations, the consonants have different pronunciations. Some of these are listed in Table 4.

| Grapheme combinations | Phoneme | Example |
|---|---|---|
| b, bb | [b], [b:] | bátur 'boat', abbi 'grandfater' |
| d, dd | [d], [d:] | dust 'dust', koddi 'pillow' |
| g, gg | [g], [g:] | gala 'to crow', sjagga 'to twaddle' |
| p, pp | [p], [Hb:] | pílur 'arrow', mappa 'folder' |
| t | [t], [Hd:] | tekja 'roof', detta 'fall' |
| k | [k], [Hg:] | kaka 'cake', krakkur 'stool' |
| f | [f], [f:] | fara 'to go', skaffa 'to provide' |
| v | [v] | øvund 'envy' |
| n, nn | [n], [n:] | nú 'now', kanna 'jug' |
| n before g or k | [N], [X] | ganga 'to walk', banka 'to knock' |
| m, mm | [m], [m:] | koma 'come', ramma 'frame' |
| l | [l] | ala 'breed' |
| r | [r][6] | læra 'to learn', marra 'nightmare' |
| h | [h] | hús 'house' |
| j | [j] | ja 'yes' |
| s, ss | [s], [ss] | siga 'to say', kassi 'box' |

Table **2**: The most frequent pronunciation of consonants in Faroese.

Palatalization occurs in Faroese of the phonemes /g, k/ in front of the front, unrounded vowels /i, e, EJ/, e.g. *geyla* 'yell' ['dZEJ:la]. The letter combinations *gj, dj, kj, tj* and *hj* also have alveopalatal sounds, e.g. *tjóvur* 'thief' ['tSOW:wUr], and when /s/ is followed *by j, kj, tj or k* followed by the aforementioned front unrounded vowels, /s/ is palatalized as well, e.g. *skip* 'ship' ['Si:b]. These are all general rules, but exceptions to the rule exist.

Table 4 shows some combinations of vowels and consonants that have an unexpected pronunciation. Note that in the list, we include the

---

[5] This is not true for the dialect of Suðuroy.

[6] /r/ can have multiple pronunciations in Faroese, including alveolar and trill. We have opted for the most frequent pronunciation, namely the alveolar.

| | Long vowel | | | Short vowel | |
|---|---|---|---|---|---|
| **Grapheme** | **Phoneme** | **Example** | **Phoneme** | **Example** | |
| *á* | [OA:] | *gráur* 'grey' | [O] | *grátt* 'grey (n.)' | |
| *a* | [EA:], [a:] (in loanwords) | *glaður* 'happy', tomat 'tomato' | [a] | *glatt* 'happy (n.)' | |
| *æ* | [EA:] | *læra* 'teach' | [a] | *lærdi* 'taught' | |
| *e* | [e:] | *meta* 'estimate' | [E] | *metti* 'estimated' | |
| *i* | [i:] | *fita* 'fatten' | [I] | *fitna* 'get fat' | |
| *y* | [i:], [y:] | *fyri* 'for, before', *myta* 'myth' | [I], [Y] | *fyrr* 'earlier', *mystiskur* 'mythical' | |
| *í* | [UJ:] | *lítil* 'small' | [UJ] | *lítli* 'small (def. m.) | |
| *ý* | [UJ:] | *sýta* 'refuse' | [UJ] | *sýtti* 'refused' | |
| *o* | [o:] | *tosa* 'talk' | [O] | *toldi* 'endured' | |
| *ó* | [OW:] | *rópa* 'yell' | [9] | *rópti* 'yelled' | |
| *u* | [u:] | *gulur* 'yellow' | [U] | *gult* 'yellow (n.)' | |
| *ú* | [3W:] | *púra* 'quite, entirely' | [Y] | *púrt* 'quite, entirely' | |
| *ø* | [2:] | *søtur* 'sweet' | [9] | *søtt* 'sweet (n.)' | |
| *ei* | [aJ:] | *heitur* 'warm' | [aJ] | *heitt* 'warm (n.)' | |
| *ey* | [EJ:] | *reyður* 'red' | [E] | *reytt* 'red (n.)' | |
| *oy* | [OJ:] | *royna* 'try' | [OJ] | *royndi* 'tried' | |

Table **3**: The different graphemes representing vowels in Faroese and their phonemic counterparts. Note that length is not represented in our G2P-dictionaries.

pronunciations relevant to the central dialect only (see Section 5 for more on dialects).

### 3.2 Some phonological considerations

Most descriptions of Faroese state that the phonemes /lmnr/ are devoiced before /ptk/ and in front of /s/ as well (e.g. Thráinsson et al., 2012). However, recent work in progress sheds doubt on this traditional description of devoicing in front of /s/. Lamhauge (2022) finds a dialectal difference between the northern and southern dialects of Faroese, suggesting that these sonorants are not as categorically devoiced as the literature describes them. However, the Ravnur project chose to phonetically transcribe the sonorants in front of /s/ as voiced sonorants.

Furthermore, when /s/ is followed by a short diphthong ending in a high vowel, i.e. the graphemes *í, ý, ei*, and *oy*, the /s/ can be pronounced as [S] (Lockwood, 2002), e.g. *píska* 'whip'. However, in this condition, /s/ is transcribed [s] in the Ravnur lexicon.

| Grapheme combination | Phoneme | Example |
|---|---|---|
| *-ógv-* | [Egv] | *krógv* 'inn' |
| *-úgv-* | [Igv] | *búgv* 'home' |
| *-ang-* | [ENg] | *svangur* 'hungry' |
| *-angi-* | [EndZI] | *svangir* 'hungry |
| *-ank-* | [ENg] | *blanka* 'polish' |
| *-eingi-* *-einki-* | [OJndZI] [OJxdZI] | *dreingir* 'boys', *einki* 'nothing' |
| *hv-* | [kv] | *hvør* 'who' |
| *-ll(-)*[7] | [dl] | *øll* 'everyone' |
| *-rn*[8] | [dn] | *bjørn* 'bear' |
| *-nn-* after *ei, oy* | [dn] | *seinni* 'later' |
| *-ðr-* | [gr] | *veðrur* 'ram' |
| *-ðg-* | [g:] | *steðga* 'stop' |
| *-ðk-* | [hk:] | *blíðka* 'make gentle' |
| *-gd-* | [d:], [gd] | *løgdu* 'laid (pl.), *løgd* 'laid (f.)' |
| *-vd-* | [d:], [Wd] | *høvd* 'head', *høvd* 'rise', resp. |
| *-rs-* | [z] | *mars* 'March' |
| *-um* (unstressed) | [Un] | monnum 'men' |

Table **4**: Grapheme combinations and their phonemic representations.

[7] In loanwords, *-ll(-)* is pronounced as [l:], e.g. *ball* 'party'.

[8] When /r+n/ are combined in the inflection of a word, it is pronounced [rn], e.g. *far+nir* (m.pl. of *farin*). There are other exceptions to the /r+n/ rule as well (Thráinsson et al. 2012).

The Ravnur project decided to make the pronunciations in the dictionary distinct, phonological pronunciations. This means that the unstressed vowels [I] and [U] are transcribed as such, even though they have merged to [8] in central Faroese spontaneous speech (Petersen, 2022). A word like *mammu* 'mother (acc.)' would therefore be pronounced [mam:8] in the Central dialect, but is transcribed as [mam:U] in the Ravnur dictionary. Also, the unstressed syllables –arnir, -arnar, -irnar and –urnar are fully phonetically transcribed, e.g. *hundarnir* 'the dogs' ['hUndarnIr], although these are frequently pronounced without an /r/ in spontaneous speech, e.g. [hUndanIr] (Adams and Petersen, 2014).

These decisions made by the original Ravnur group have been followed in both our CENTRAL and EAST dictionary. Research remains to show how these phonological conditions are actually produced, and whether or not there is dialectal difference.

## 4 Representing pronunciation variations in one form

For developing the G2P-tool, we used the open source pronunciation dictionary from project Ravnur (Debess et al., 2019; Simonsen et al., 2022). However, we only use one single pronunciation per word. The original dictionary had many entries with multiple pronunciations, and for this project of developing the G2P-tool we had to prioritize just one of them. Being a descriptive dictionary, some words were assigned multiple pronunciations due to language variation (differences in dialect, sociolect or other lects) and assimilation of different kinds. In cases of dialectal variations, the pronunciations of the central dialect were chosen, as the different dialects are represented through separate dictionaries. In cases of other variation with no research to base the decision on, the choice was based on (by the Faroese linguist) perceived frequency of the forms, choosing the more frequent as the primary - knowing the limitations of this method.

Quite many of the word forms with multiple pronunciations were due to homographs. As the dictionary version for the G2P tool only operates with the values of orthography and pronunciation and no grammatical or semantic information, homographic word forms that belong to different lexemes melt into one, with their respective pronunciations being registered as pronunciation variations of the same form. As this tool strictly focuses on the relationship between graphs and phones, the semantics and grammar to differentiate homographic word forms are of course obsolete, but the omission of these values also presents challenges. Choosing one pronunciation over another in these cases, where both forms are considered to be valid in Faroese language, blurs the depiction of the linguistic reality, but is necessary for this linear conversion tool and might even increase accuracy of the tool.

In cases of multiple pronunciations due to homographs, one pronunciation has been chosen to be primary based on the main criteria of frequency and second phonological heritage (detailed below), having functionality and error rates of the tool in mind.

Frequency derived from searches in available corpora or web-search-based frequency was not always sufficient due to the relatively small volume of the resources, many of them not tagged, making it difficult to distinguish homographic forms. In these cases, other frequency measures were also taken into account:

- Native speaker intuition. Example: *havi* 'to have' PRES.1.SG /hEAvI/ > *havi* 'garden' SG.NOM /ha:vI/.
- Function words > content words.
- Grammatical case of the word form (the genitive is very rare (Thráinsson et al., 2012), and the pronunciation pairs were often due to a genitive word form opposite a word form with another grammatical case or from another part of speech). Example: *loksins* 'finally' (adverb) /lOgsIns/ > *loksins* 'lid' SG.GEN.DEF /lo:gsIns/.
- Frequent > obscure inflectional forms.

Though frequency being the main criteria, we also took into account and implemented rules of phonological heritage:

- Words of linguistic heritage > loan words (even though the loan word is well implemented in Faroese). This ensures the systematic phonological rules of Faroese a broader representation in the data. Example: *banki* 'to knock' PRES.1.SG /bExdZI/ > *banki* 'bank' SG.NOM /baxdZI/
- Common nouns > proper nouns. Proper nouns in general follow the phonological

rules to a lesser extent than common nouns. Examples: *allan* 'all' SG.MASC.ACC.DEF [adlan] > *Allan* 'Alan' (person name) [alan].

## 5 Two dictionaries for Faroese

The original dictionary from the Ravnur project is based on the dialect of the capital in the Faroe Islands. Even though there is no official standard dialect for Faroese (Petersen, 2022), there is believed to be some kind of central Faroese based partly on the dialect of the capital and partly on the written language (Jacobsen, 2011; Knooihuizen, 2014). However, in working with this G2P tool, we wanted to have greater diversity and decided to make two versions of the original dictionary reflecting two different dialect areas. The two dialect areas are the central dialect area, where the capital is located, called CENTRAL henceforth, and part of the northwest dialect area, called EAST henceforth (see Figure 1). The CENTRAL dialect area has the largest number of inhabitants, and the EAST dialect area has the second highest number of inhabitants. Combined, these two dialect areas comprise around 71% of the population[9].

The islands in the northwest area are for several reasons classified as being the same dialect area in the most recent dialect classification (Petersen, 2022). However, there is one important phonetic difference between the westernmost islands and the more central and eastern islands in that dialect area, and therefore, the westernmost part of the dialect area is not included in our EAST dictionary (the dotted line in Figure 1 marks the two parts of the northwest dialect area). This phonetic difference is the pronunciation of the digraph *ei*. For the same reason, we have given this dictionary the name EAST. This way, it is possible to make WEST, NORTHERN and SOUTHERN dictionaries as well, should the possibility present itself.

In Section 3, we presented the general phonological rules for (central) Faroese. In the following section, we will outline the main dialectal differences between the CENTRAL and the EAST dictionaries. For further information on dialectal differences in Faroese, see Thráinson et al. (2012) and Petersen (2022).

### 5.1 Phonological differences between CENTRAL and EAST

The main dialectal differences between the CENTRAL and EAST dialect area are as follows:

1) The letter *ó* is pronounced [OW] in the central dialect area and as [9W] in the east dialect area, 2) the digraph *ei* is pronounced [aJ] in the central area and as [OJ] in the east area, and 3) lack of preaspiration after long, non-high vowels before fortis[10] stop closures in the central area and preaspirated stop closures in the same condition in the east area (Petersen, 2022).[11] We will go through each of these three features in turn and explain how we did the changes, exceptions to the rule etc.

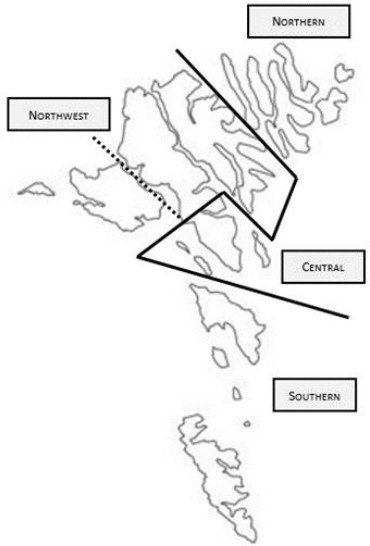

Figure **1**: The four main dialect areas based on Petersen (2022). The area to the west of the dotted line is the area from the northwest dialect area that is not part of our EAST dictionary.

### Variation in pronunciation of *ó*

The CENTRAL dictionary being the starting point for creating an EAST dictionary, we needed to convert all [OW] sequences in the dictionary to [9W] sequences. The conversion was straight forward. Exemptions to this conversion were 1)

---

[9] https://hagstova.fo/fo/folk/folkatal/folkatal

[10] The terms 'fortis' and 'lenis' are often used on an abstract level to distinguish between the stop series /ptk/ and /bdg/ (Helgason 2002; Hejná 2015; among others). In using these terms, we are simply following this tradition, not implying any specific phonetic differences between the two series.

Others call these series hard and soft, respectively (Thráinsson et al. 2012).

[11] As the phonetics of Faroese in general are understudied (Lamhauge 2022), there might be other dialectal differences that have not been described or studied yet, as is shown in Lamhauge (2022) in the case of sonorant devoicing. In cases of doubt, we have followed the decisions made by the original Ravnur project.

loan words and 2) words with the orthographic sequence -ov-. Loan words or foreign words (especially from English), which have not been implemented enough in Faroese language to adapt to dialectal differences in pronunciation, were not converted, e.g. Windows [vIndOWs] (proper noun) and karaoke [karaOWki].

The letter sequence *-ov-* can also manifest in an [OW]-pronunciation in Faroese, e.g. in *bakarovnur* 'oven' SG.NOM [bEAgarOWnUr] and *flovislig* 'embarrassing' SG.FEM.NOM [flOWwIsli]. The -ov- based [OW] has no dialectal variation and is pronounced the same throughout all the dialects. In the conversion process of [OW] to [9W], words with the orthographic sequence *-ov-* together with an [OW] in pronunciation were therefore exempt, and their transcriptions were not converted.

The pronunciation of phonologically short *ó* is not relevant for distinguishing CENTRAL and EAST and is not discussed further here.

**Variation in pronunciation of *ei***

The pronunciation of the phonologically long digraph *ei* is dialectally distributed as mentioned in the previous section. We converted all [aJ] sequences in CENTRAL to [OJ] sequences in EAST. Exemptions to this conversion were 1) loan words and 2) proper nouns. Loan words or foreign words (especially from English), which have not been implemented enough in Faroese language to adapt to dialectal differences in pronunciation were not converted. This applies to loan words, e.g. *gurkumeia* 'turmeric' SG.NOM [gUzgUmaJja] or *bei* 'bye' [baJ][12]. Proper nouns with long *ei* behave quite differently regarding pronunciation than other parts of speech, and even though there is a variation in the pronunciation (between [aJ] and [OJ]), the variation seems to be non-systematic and idiolectal. For this reason, proper nouns were exempt from the conversion process. The exemptions of loan words and proper nouns were done manually.

**Variation in preaspiration**

Table 5 shows an overview of the phonological conditions in which preaspiration occurs in different Faroese dialects. In some phonological conditions, preaspiration occurs in all the dialects, i.e. between a short vowel and a long stop closure, or after a short vowel and before a stop closure followed by a sonorant. However, between a long, non-high vowel and a short consonant, only some dialects preaspirate. If the vowel is high in this condition, none of the dialects preaspirate (Thráinsson et al., 2012). There are other phonological conditions as well, in which preaspiration might or might not occur. However, as these have not been studied sufficiently, they are not included in this overview.[13]

| Phonological condition | Example | Dialects |
|---|---|---|
| VC:(ptk) | *Átta* 'eight' [OHd:a] | All dialects |
| VC(ptk)C(mn) VC(t)C(l) | *Vatn* 'water' [vaHdn] | All dialects |
| V:(non-high)C | *Kaka* 'cake' [kEA:Hga] | Only some dialects (including EAST, excluding CENTRAL) |

Table **5**: An overview of the phonological conditions in which preaspiration occurs in different Faroese dialects (based on Thráinsson et al., 2012).

This means that in the CENTRAL dictionary, there is no preaspiration in the V:C condition. We therefore had to insert preaspiration in this condition in the EAST dictionary. As the CENTRAL dictionary does not have length marks neither on vowels nor on consonants, the process could not be done automatically. We searched for all of the relevant phonological conditions, but as there are quite a few exceptions to the rules, i.e. in loanwords, all of the instances had to be checked manually before implementing the change. Furthermore, as there is actually not much known about the pronunciation of stops in loanwords, we have made educated guesses based on native speaker intuitions at times. For example,

---

[12] Words with the [aJ] pronunciation can also be found with other spelling than *ei*, e.g. *wifi* 'wifi' [vaJfaJ] and *kai* 'dock' SG.NOM [kaJ], but this is due to the foreign heritage of the words. All words with [aJ] and other spelling than *ei* were also exempt.

[13] For example, Lamhauge (2022) has found in her work in progress a dialectal split in the pronunciation of non-

homorganic stops. In these cases, we have followed decisions made by the original Ravnur project, and we have made no changes to these decisions in the EAST dictionary. See Thráinsson et al. (2012) and Helgason (2002) for a more thorough overview of preaspiration in Faroese.

in loanwords that end in –at, we have chosen to insert preaspiration, e.g. salat 'salad' ['salaHd]. In other cases, the decision was made on a word basis.

# 6 G2P

The "Faroese G2P Models" is a set of two models trained with the software tool "Sequitur-G2P" (for details on Sequitur-G2P, see Bisani and Ney, 2008), which is a trainable grapheme-to-phoneme tool developed at RWTH Aachen University (https://www.rwth-aachen.de/) by Maximilian Bisani. One of the models is for Central Faroese and the other is for the East variant.

The set also includes two files with the corresponding repertory of phonemes for the Central (central.phones) and the East (east.phones) variants.

## 6.1 The training set

In order to train the Sequitur-G2P, it is necessary to provide it with a pronunciation dictionary that will play the role of training set. This pronunciation dictionary was taken from BLARK 1.0 and double-checked by experts and finally split into one set for the Central and East variants of Faroese. The characteristics of both training dictionaries are the following:

- Both dictionaries contain 197,757 unique words each.
- Words with symbols other than letters of the Faroese alphabet (e.g. *kt-vinnu*, *stóra_dímun*) were excluded from the training dictionaries because those symbols do not have a correspondence in phonemes.
- The pronunciations are based on the FARSAMPA alphabet provided in the BLARK 1.0.
- Multiple pronunciations for one particular word are not accepted. Therefore, there is only one pronunciation associated with each word. This is to avoid providing Sequitur with inconsistencies (see more in Section 5)[14].
- The number of phonemes for the Central variant is 60, which is a subset of the East variant, which has 63.

Most of the entries in both dictionaries are the same, the only difference occurs with words that can be pronounced differently in both variants.

As the Sequitur models are destined to do ASR experiments, the diacritics for length (:), primary stress (%), secondary stress (~) and emphasis (!) were not taken into account, because we saw that they do not offer any advantage to the ASR experiments but they make the models unnecessarily more complex instead[15].

## 6.2 Evaluation and results

In order to evaluate the performance of the Sequitur models, a set of 1000 words with pronunciation was randomly selected for each variant. The resulting test sets do not contain the same words, and the test sets are not included in the training dictionaries.

The evaluation was performed using the evaluation command provided by Sequitur. Table 1 shows a summary of the results obtained:

|  | **Central Model** | **East model** |
|---|---|---|
| Total | 1000 strings, 9703 Symbols | 1000 strings, 9615 symbols |
| Successfully translated | 100% strings, 100% symbols | 100% strings, 100% symbols |
| string errors | 22 (2.20%) | 31 (3.10%) |
| symbol errors | 29 (0.30%) | 44 (0.46%) |
| insertions | 7 (0.07%) | 4 (0.04%) |
| deletions | 3 (0.03%) | 5 (0.05%) |
| substitutions | 19 (0.20%) | 35 (0.36%) |
| translation failed | 0% strings, 0% symbols | 0% strings, 0% symbols |
| total string errors | 22 (2.20%) | 31 (3.10%) |
| total symbol errors | 29 (0.30%) | 44 (0.46%) |

Table **8**: Evaluation results obtained from Sequitur's evaluation command

As can be seen in Table 8, the translation errors are below 5% in both models, indicating that the models are reliable and not far away from other models found in the literature (Milde et al., 2017).

---

[14] This is also explains why we trained one model for each dialect, instead of training one joint model and apply rules as a post-processing step.

[15] In the case of TTS, it would be beneficial to train a different model for that purpose that includes the diacritics.

# 7  Conclusion

A standardized pronunciation dictionary of good quality is crucial for the development of spoken language technologies. This work describes the definition, development and the establishment of a Faroese pronunciation dictionary and a grapheme-to-phoneme tool to go along with it. This work is also important in understanding spoken Faroese and can be used to study regional differences in accents and dialects. It is clear that this will form a basis of further studies of Faroese and development of spoken language technologies such as ASR and TTS.

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
