# OpenReview forum: "Standardising Pronunciation for a Grapheme-to-Phoneme Converter for Faroese"
_NoDaLiDa/2023/Conference — NoDaLiDa 2023_

### Official Review · Reviewer_NCW1 · 2023-03-12
**An importam resource for the development of spoken language technologies for Faroese**

**Rating:** 7
**Confidence:** 4

**Review:**

The article describes in details Faroese language SAMPA development, outlines some of the most common grapheme-to-phoneme conversions of Faroese, and development of pronunciation dictionaries that is used in the G2P model developed by the authors of the articles.
One can only agree that a standardized pronunciation dictionary of good quality is crucial for the development of spoken language technologies. Described solution to represent pronunciation variations in one form seems to be successful.
The reasons  why pronunciation dictionaries of Central and East Faroese were chosen do not seem convincing.


**Paper Type:**

Long paper

---

### Official Review · Reviewer_eqEA · 2023-03-14
**Creation of the first Faroese SAMPA alphabet and the training and evaluation of two grapheme-to-phoneme models**

**Rating:** 7
**Confidence:** 3

**Review:**

This paper both reports on the creation of the first Faroese SAMPA alphabet and the training and evaluation of two grapheme-to-phoneme models.

I enjoyed reading this paper a lot. The creation of the SAMPA alphabet is well described, and evaluating it by using for transcribing the words in the Ravnur lexicon seems like a very sensible way of ensuring that it is complete. Also the training of the grapheme-to-phoneme models were clearly described, evaluated and yielded good results. In general, new resources for under-resourced languages are very welcome (and under-resourced languages in the Nordics is escpecially suited to Nodalida), so this, in combination with it being a well-written paper with solid experiments makes it a good fit for the conference.

Some comments:

- "Postaspiration and other relevant diacritics for Faroese (e.g. voiced/unvoiced) are integrated in the character system and not denoted via symbols of their own." If you have space: Could you give a concrete example here?
- "Frequency: with a relatively small foundation for search-based frequency". What do you mean by search-based frequency? Do you search in a corpus? In that case which one? Or do you mean web search?
- You use a rule-based method for converting your training data from the central dialect to the east dialect. And then you train two G2P models, one for the central dialect and one for the east. Wouldn't it also be possible to just train one model and apply the rules you have used as a post-processing step? I'm not meaning that this would be better, I'm just wondering if you could comment on that possibility, and whether it would be possible.
- "As the Sequitur models are destined to do ASR experiments, the diacritics for length (:), primary stress (%), secondary stress (~) and emphasis (!) were not taken into account, because we saw that they do not offer any advantage to the ASR experiments". Wouldn't this be a disadvantage if you are using the models for text-to-speech? Would you train a different model for that? Could you please comment on that.
- It would be good with a short text that says something more about the Sequitur models. What kind of techniques is it based on?

**Paper Type:**

Long paper

---

### Decision · Program_Chairs · 2023-03-17

Accept